# Optimisation of an Industrial Optical Sorter of Legumes for Gluten-Free Production Using Hyperspectral Imaging Techniques

**DOI:** 10.3390/foods13030404

**Published:** 2024-01-26

**Authors:** Roberto Romaniello, Antonietta Eliana Barrasso, Claudio Perone, Antonia Tamborrino, Antonio Berardi, Alessandro Leone

**Affiliations:** 1Department of Agriculture, Food, Natural Resource and Engineering, University of Foggia, 71122 Foggia, Italy; roberto.romaniello@unifg.it (R.R.); antonietta.barrasso@unifg.it (A.E.B.); claudio.perone@unifg.it (C.P.); 2Department of Soil, Plant and Food Science (DISSPA), University of Bari Aldo Moro, Via Amendola 165/a, 70126 Bari, Italy; antonia.tamborrino@uniba.it (A.T.); alessandro.leone@uniba.it (A.L.)

**Keywords:** hyperspectral imaging, optical sorter, gluten-free, legumes

## Abstract

The market demand for gluten-free food is increasing due to the growing gluten sensitivity and coeliac disease (CD) in the population. The market requires grass-free cereals to produce gluten-free food. This requires sorting methods that guarantee the perfect separation of gluten contaminants from the legumes. The objective of the research was the development of an optical sorting system based on hyperspectral image processing, capable of identifying the spectral characteristics of the products under investigation to obtain a statistical classifier capable of enabling the total elimination of contaminants. The construction of the statistical classifier yielded excellent results, with a 100% correct classification rate of the contaminants. Tests conducted subsequently on an industrial optical sorter validated the result of the preliminary tests. In fact, the application of the developed classifier was able to correctly select the contaminants from the mass of legumes with a correct classification percentage of 100%. A small proportion of legumes was misclassified as contaminants, but this did not affect the scope of the work. Further studies will aim to reduce even this small share of waste with investigations into optimising the seed transport systems of the optical sorter.

## 1. Introduction

Due to the growing interest of buyers in the area of health and food safety, there has been an increase in information about the food they buy on a daily basis. Particular attention has been paid to products that are considered niche products, including gluten-free products, aimed at an ever-increasing segment of the population. These kinds of products cover a large interest to overcome the coeliac disease that is a systemic and chronic autoimmune disease that develops in predisposed individuals, following the intake of gluten proteins present in products such as wheat, barley and rye. It is a disease that has seen an increase in incidence in individuals of all ages in recent decades, detected using state-of-the-art diagnostic tests [1].

The aim of the analysis conducted is to identify methods to discriminate gluten-containing contaminants in gluten-free products, to lower the quantities present in these foods.

The composition of products destined for this category, to date, cannot be totally deprived of gluten, which, in small quantities, can be consumed by coeliac individuals. In a gluten-free diet, the daily amount not to be exceeded is 10 mg. Despite this, it has been shown that both natural and certified gluten-free foods can be heavily contaminated with gluten well above the commonly accepted threshold of 20 mg/kg. Studies aim to lower this threshold, on the one hand, making diets more in line with consumer need and health, and on the other hand, making consumers more aware and informed about the limits and overall amounts of gluten in diets [2].

Currently, the removal of these contaminants is carried out along the production line by means of mechanical separation, which consists of two main steps, namely, the removal of extraneous materials and their classification on a size basis [3].

Machine vision inspection systems are currently used in many pickle processing facilities, but they are designed for inspecting external characteristics (size, shape, and colour) and/or surface defect and are inappropriate for detecting internal defects [4].

The parameters used do not allow for total separation, since the contaminants are not homogeneous with each other and do not always differ greatly from the production we want to separate; likewise, the efficiency increases with the degree of homogeneity of the final product. In the case presented, we consider legumes for gluten-free production, which differ from each other in shape, size, volume and colouring, making the identification of normalized parameters complicated.

Over the years, several studies have been conducted on the detection of gluten in products. A group of researchers has structured a technique for detecting gluten in food, based on fluorescence spectroscopy. The new technique allows for the detection sensitivity of 0.006 ppm of gliadins, the main protein responsible for coeliac disease, whereas the current techniques on the market reach 32 ppm [5].

This is a destructive method that does not allow for the entire production to be analysed, as it involves dissolving a food sample in a cocktail of enzymes mixed with a concentrated solution of gliadins labelled with a fluorescent molecule and then adding the IgG isolated from the mice. This technique is, moreover, not very feasible due to the need for many expensive and scarce media. No way has been found yet to miniaturise the fluorescence assay so that it can also be used by unskilled individuals.

Innovation in food plays a very important role in overcoming these kinds of problems. In particular, automation systems, which have been developed thanks to simulation studies on food plants [6], are efficient in controlling the entire production process. Concerning the automation of food plants, in the past few decades, the winning technique for the detection of food contaminants turned out to be hyperspectral imaging [7]. Thanks to these features, food plants can be automated and controlled in real time. It is very popular in the food industry due to its speed [8], objectivity and low cost [9,10,11,12,13,14]. It also has the advantage of being a non-destructive technique that allows for the entire production to be analysed [15,16]. This technique correlates spectroscopy and imaging, providing information about the chemical properties of a material and their spatial distribution [17] to identify [18], quantify and classify small details of the object under investigation, as well as internal physical and chemical properties of the same object [19,20,21].

The objective of the research, therefore, was to identify the reflectance spectra in the visible and near-infrared (VIS/NIR) spectrum of leguminous products and gluten-containing contaminants in order to create a statistical classifier capable of accurately discriminating between the two product types. A methodology based on hyperspectral image analysis and chemometric techniques was applied to construct the statistical classifier. The statistical classifier was then tested on an industrial optical calibrator and the separation performances of the contaminant.

## 2. Materials and Methods

The experimental tests were conducted in two steps. The first involved analysing the hyperspectral data of each product in the way of identifying influential wavelengths in order to classify, in pairs, a legume with a gluten-based contaminant. A statistical classifier based on a neural network was used for this purpose. In particular, the linear classifier was used. The classifier is able to test and validate the classification function by prioritising the best predictors (in this case, the most influential wavelengths). The aim of the work is to identify classification models that can allow for the almost-total exclusion of gluten contaminants from legumes. Therefore, the number of predictors was chosen based on obtaining a classification rate above 99%. The second step of the experiment was to test the determined classification models directly on an industrial optical calibration machine equipped with VIS/NIR sensors in the 400–1700 nm wavelength range.

### 2.1. Samples

Three types of legumes and three types of cereals were considered for the survey. Legumes were broad beans, chickpeas, and lentils, while cereals were wheat and barley. The choice of these products was dictated by the type of crops grown in Italy and the crop rotations that are usually carried out were also considered (Figure 1).

Ten kilos of legumes for each type considered and ten kilos of gluten contaminants, again for each of the three types considered, were subjected to hyperspectral investigation. The legumes were sourced from a producer company in the region of Apulia (Italy). The legumes were randomly sampled within a large production lot in order to collect physical and dimensional variability. Within these batches of 10 kg each, intact and non-intact product units were present in order to simulate the real operating condition of the sorting stations. Cereals, on the other hand, were sampled from silos on the farm.

### 2.2. Hyperspectral Imaging System

In order to investigate the optical properties of legumes and contaminants, an analysis protocol based on hyperspectral imaging was implemented. The instrumentation used was a hyperspectral plane scanner DV-Optic s.r.l., equipped with two types of sensors. The first is of the VIS/NIR type, with a Specim Impsector V10 spectrometer (Specim spectral imaging LTD, Oulu, Finland), with a wavelength range from 400 to 1000 nm, with a sampling step of 5 nm, while the second, of the SWIR (Short Wave Infra-Red) type, features an InGaAs sensor (Xeva InGaAs VIS/NIR 320 × 256 FPA, Xenics nv, Leuven, Belgium), capable of acquiring reflectance at wavelengths from 1000 to 1700 nm, with a sampling step of 5 nm. The illumination sources are different for the two types of sensors. For the VIS/NIR, a halogen illuminant (150 W) was used, to which a fibre optic guide terminating in a linear collimator was connected in order to illuminate the scanner’s line of sight. For the SWIR sensor, a halogen source with an emission spectrum between 800 and 2000 nm was used (EKE 9596ER, 150 W). This illuminant was also equipped with a fibre-optic guide and linear collimator (Figure 2). To enable the sensors to receive the reflected radiation from the object, two mirrors oriented at 45° were placed in the line of sight of the two sensors (Figure 2b).

The spectral data were analysed with the Statistics and neural network toolbox in Matlab r2023b (The Matworks, Natik, MA, USA). Reflectance spectra were normalised using the following method [22]:(1)$$\;R_{C\lambda }=\frac{R_{\lambda }-D_{\lambda }}{W_{\lambda }-D_{\lambda }}$$
where R_λ_ is the recorded hyperspectral image, D_λ_ is the dark component image (0% reflectance) acquired by turning off the lighting source with the lens of the camera completely closed by a black cap, and W_λ_ is the white reference component image (acquiring the white board reference). Equation (1) was iteratively applied for all 261 component images to obtain the corrected component images (Rc_λ_). These corrected images were repacked into a hyperspectral image. The hyperspectral images were then exported in an ENVI format. Using MATLAB r2023b software, the hyperspectral images were read and processed, removing the acquisition background and calculating the average reflectance value, at all wavelengths, of the image areas characterised by the product units. Figure 2c shows an example image, in pseudo colours, of the masked hyperspectral image of broad beans samples. The hyperspectral data were then stored in vectors in order to generate the relative reflectance spectra (Figure 2d). The reflectance spectra obtained were derived from the analysis of numerous hyperspectral images for each product type considered. Figure 2d shows the 30 reflectance spectra for the 30 product units in the hyperspectral image in Figure 2c.

### 2.3. Data Analysis

A classification function was developed to classify legumes against gluten contaminants with a high degree of accuracy. For this purpose, datasets were created containing the paired spectral data of a legume and a contaminate. Thus, 6 datasets were considered (2 legumes × 3 contaminates). Statistical classifiers were tested using Matlab functions. For each data set, 9 types of classifiers were tested. Each classifier type includes some variants (i.e., linear, quadratic, gaussian, etc.). In Table 1, the value of correct classification rate refers to the best value obtained for each classifier type. This preliminary analysis shows the SVM classifier as the best one for the datasets considered. In particular, the best sub-category resulted from the support vector machine-linear classifier (SVM-LC).

Therefore, once the SVM-LC was selected, a training set and a validation set were prepared for each dataset by selecting 50% of original data for the training and 25% + 25% for validation and testing. A 5-fold cross-valuation method was used to validate the classification model. As the aim of the work is to eliminate gluten contaminants from the legumes, the classifier was trained and tested to achieve correct classification rates close to 100%. The performance of the classifier was evaluated by measuring the percentage of the correct classification obtained (CCR), the percentage of false positives (FPR) and false negatives (FNR), using validation and test data. In particular, true positives (TPR) pertain to legumes correctly classified as legumes, while false positives pertain to contaminants incorrectly classified as legumes. Regarding the aim of the research, it is important that there are no contaminants in the selected legumes, so the desirable situation would be to have a very high CCR and an absence, or very low value, of false positive rates (FPRs), resulting from the subtraction between TPR and FNR. The presence of false negatives (thus, the absence of false positives) avoids the presence of contaminants in the legumes, but a high value of false negatives (FNR) would result in a loss of legumes in the waste, resulting in economic damage.

To identify the most influent wavelengths to classify the two classes of interests (legumes and gluten contaminants) on the basis of reflectance data, a method for wavelength selection was used. In particular, the large dataset Ω (261 wavelength) obtained from hyperspectral imaging system was subjected to the Feature Selection Minimum Redundancy Maximum Relevance (FS-MRMR) algorithm. The method uses a forward addition scheme to populate an empty set, S, containing the best wavelength to be used for the mathematical model. For this purpose, the algorithm uses a function of interdependence (I) [23,24], indicating the interdependence or not of a couple of λ, and a mutual information quotient (MIQ) value to rank features (wavelengths). The MIQ (Equation (2)) includes two indexes, $V_{\lambda }$ and $W_{\lambda }$, indicating the relevance and redundance of λ with respect to the response variable y (product class: legumes or gluten contaminant).
(2)$$\;MIQ=\frac{V_{\lambda }}{W_{\lambda }}$$
where
(3)$$V_{\lambda }=I(\lambda ,\;y)$$
(4)$$W_{\lambda }=\frac{1}{\left|s\right|}=\sum_{zeS}I(\lambda ,\;z)$$

A simplified scheme of the FS-MRMR algorithm is depicted in Figure 3.

Having defined the set S, containing the most representative wavelengths, the classification function was implemented for each dataset, to classify the data based on the selected wavelengths. 

### 2.4. Industrial Hyperspectral Sorter

The sorting machine (Figure 4a) consists of five independent channels for product selection. Each channel can make a specific type of selection based on what is programmed by the PLC. In the figure the selection process is highlighted by different colours ball separated in two different final boxes.

The loading of the product as it is takes place from above into continuously fed hoppers.

In the sorter the product is made to slide in free fall and along a slide and in a space of about 10 cm, it is analysed with n. 2 near infrared camera with scan rate of up to 15,000 Hz and optical resolution of 0.06 mm (60 µm) and n. 2 RGB full-colour cameras (front and back) with 4096 pixels working in the visible spectrum and the inspection system recognizes 16 million colours that combined to 0.06 mm optical resolution that detect the parameters set in the selection protocol loaded into the PLC of the machine and reject the products identified by the software using jets of compressed air controlled by solenoid valves.

In Figure 4b,c, products that do not comply with the selection requirements are highlighted in purple and with red boxes.

For the industrial tests, 30 kg of each type of legume considered for laboratory analysis was used. The three types of legumes were polluted with known doses of wheat and oats in order to verify the selection capacity. In addition to cereals, stones were included as an additional contaminant likely to be present. 

The PLC operating the sorting machine was programmed by entering the algorithm compiled on the basis of the statistical classifier developed in the MATLAB environment. For the selection of the wavelengths identified in the laboratory experiment, it was decided to use the broadband mode. In this way, the software of the sorting machine uses a bandwidth of 15 nm, instead of 5, in order to encompass the narrowband wavelengths identified for the legume–grain and legume–oat discrimination. The test phase included three successive passes of the same product in order to verify the robustness of the measurement by the optical system. The results of the three replications were subjected to statistical analysis (one-way ANOVA and Tuckey test) to check whether there were any deviations. The statistical analysis was performed with the Statistic and Machine Learning toolbox of MATLAB. 

## 3. Results

### 3.1. Laboratory Tests

Tests conducted in the laboratory resulted in a classification function for each dataset considered, capable of classifying legumes from gluten contaminants with a high percentage of correct classification. In particular, the FS-MRMR algorithm for the selection of influential variables was very efficient. As shown in Table 2, the algorithm on the basis of the dataset assigned a score to each wavelength. 

Starting with the variable with the highest score, the algorithm incrementally selected subsequent wavelengths until the highest percentage of correct classification was reached. The selected wavelengths then differed in both type and number based on the pairs considered. Based on the validation dataset (25% of total data), the most restrictive situation was observed for the chickpea–oats pair. In this case, the percentage of correct classification was not 100% as in the other cases but stood at 99.2%. On the other hand, it must be said that the 0.8% error (false negative rate) concerned a portion of legumes incorrectly classified as contaminants. Therefore, despite the classification error, the selected legume product was still found to be contaminant-free. Thus, the error resulted in only a small percentage of the legume being discarded as a contaminant.

After the validation procedure of the classifier, it was tested with the testing set (25% of the total data). These data are unknown to the classifier, so the testing procedure is used to assess the generalization capability of the classifier. Table 3 shows the test results.

The test data showed a high generalization capacity of the statistical classifier. As shown in Table 3, all datasets showed an FNR value below 1% and no FPRs were recorded. This result is very important because a false positive (FPR) represents a product unit classified as a ‘legume’ but is a ‘contaminant’. Since no FPRs were recorded, the system is certainly capable of discarding all contaminants, i.e., not including a unit of contaminant in the mass of selected legumes.

These results refer to laboratory analyses with acquisitions performed under optimal conditions. The next section will discuss the results of tests on the industrial optical sorting machine where operating conditions are more restrictive, with shorter acquisition and processing times.

### 3.2. Industrial Tests

Tests dedicated on beans, lentils and chickpeas had been conducted by industrial optical sorter. For each product, the tests were repeated five times with contaminated product processed by an optical sorter, and results in terms of CCR, TPR, FPR and FNR are registered in Table 4. It should be noted that, in this test, each of the three legume types (broad bean, lentil, chickpea) was polluted with all two contaminants (oat, wheat). This choice was dictated by the fact that we wanted to test the ability of the optical sorter to effectively separate contaminants in general. Thus, the grading function in this case performs the analysis on the two contaminants simultaneously, unlike in the preliminary laboratory analysis where each legume was discriminated separately from each of the two contaminants.

The analysis in Table 4 shows the performance of the statistical classifier developed, as well as the capabilities of the electronics of the optical sorter.

Firstly, it should be noted that as in the laboratory tests, no FPRs were recorded. This is an important first result, since, as explained in the previous section, the sorter was able to correctly separate the contaminants, relegating them to the appropriate discharge channel. It is reiterated that had an FPR been recorded, the machine would have considered a contaminant as ‘legume’ and the latter would have ended up in the legume collection vessel, contaminating it. Another relevant finding was that of the TPR, i.e., the proportion of ‘legumes’ correctly classified as ‘legumes’. It can be seen that the percentage of TPR is very high, greater than 99%. Since there is no FPR, the difference to 100 of the CCR value is represented by FNR (false positives). In this case, an average value ranging from 0.33% (lentil contaminant selection) to 0.64% (chickpea contaminant selection) of ‘legume’ was classified by the system.

## 4. Conclusions

A mechanical selection, for industrial use, to separate legumes from contaminants with percentages suitable for the classification of the products as gluten-free can only take place through the use of optical sorters with n. 2 near infrared cameras and n. 2 RGB cameras, which allows for a scan of the product over the entire surface.

The analysis with infrared and RGB technology allows for analysing different parameters of the product to be selected which concern the geometry (size and shape), the colour and type of material.

The tests performed show a high percentage of true positives, i.e., legumes correctly classified as legumes, and a value of 0% of false positive rates (legumes incorrectly classified as contaminants). The value, albeit minimal, of false negative rates indicates a loss of legumes in the waste which could only be improved with further research in the selection sector. This problem is probably attributable to the speed of the product flowing through the sensors and the orientation of the product. Although the electronics are fast in both acquisition and discrimination, it is possible that during analysis, some product units may position themselves in the profile, not exposing the surface to the optics. This aspect can be considered in further research, e.g., by improving the product conveying system to the acquisition station, providing a singularization system that can allow for correct positioning, especially for larger products, when falling towards the sensors. In the tests carried out on the industrial sorting machine, the factory settings were used, whereby the falling speed of the seeds was automatically adjusted based on the acquisition speed of the optical systems. The next step could be to compile software capable of interacting with the electronics of the sorting machine so as to be able to intervene with the acquisition speed and therefore the speed at which the seeds fall. In this way, it will be possible to evaluate the selection efficiency based on the process speed. The speed variation could involve an adjustment of the geometry of the conveyor and, also in this case, make it possible to intervene with mechanical modifications to the section of the conveyor downstream of the product unloading hopper.

## Figures and Tables

**Figure 1 foods-13-00404-f001:**
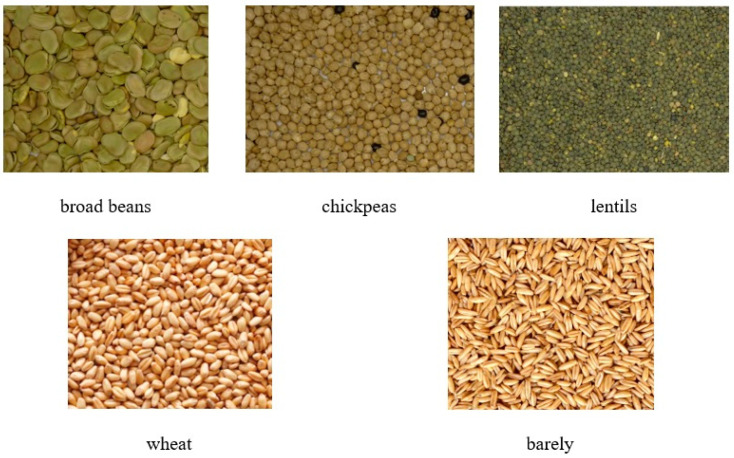
Legumes (first row) and cereals (second row) for the trials.

**Figure 2 foods-13-00404-f002:**
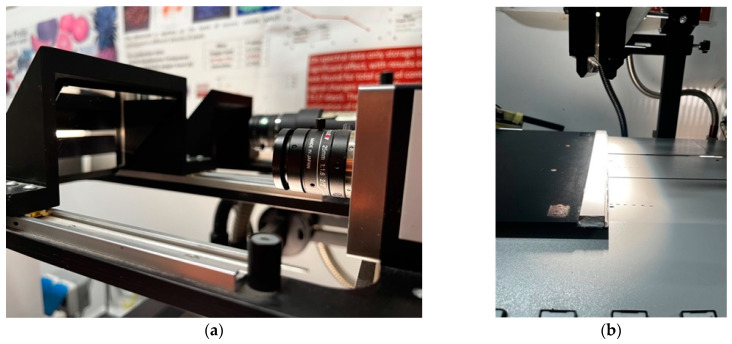
Hyperspectral imaging system: (**a**) detail of the two sensors and the 45° mirrors, (**b**) detail of fiber optics and light collimators, (**c**) pseudo image of VIS/NIR cameras, (**d**) spectra of the segmented objects in image (**c**).

**Figure 3 foods-13-00404-f003:**
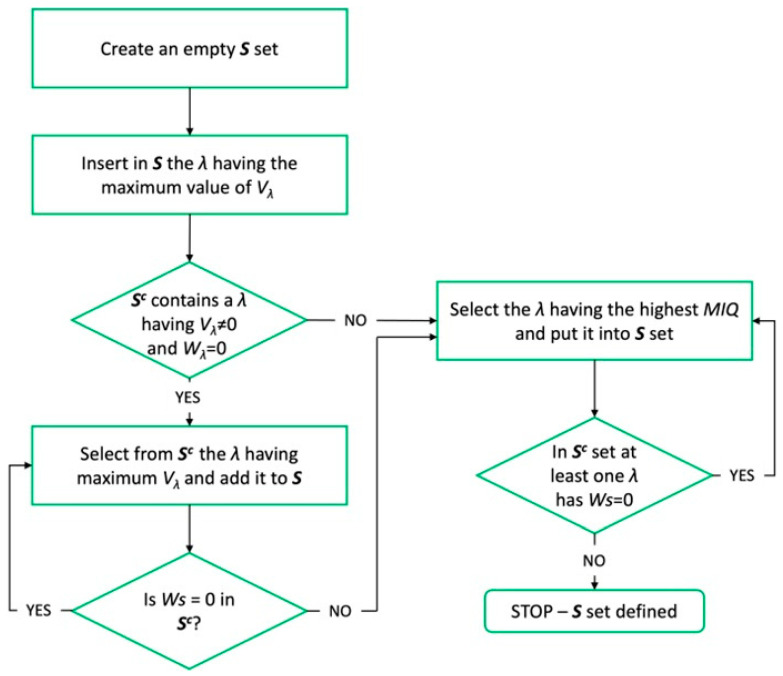
FS-MRMR algorithm flux diagram. Λ = wavelength range 400–1000, 5 nm step; $V_{\lambda }$ = relevance of the selected wavelength; $W_{\lambda }$ = redundance of selected wavelength; Sc = (Ω − S), indicating the complement set of S.

**Figure 4 foods-13-00404-f004:**
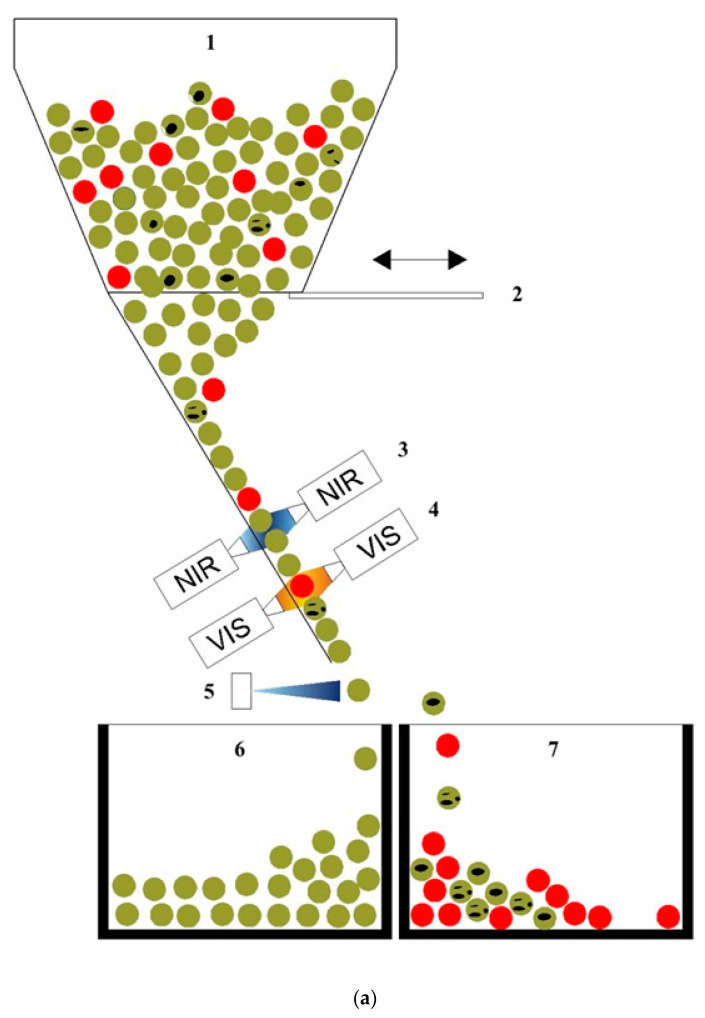
Automatic industrial sorter. (**a**) Operative scheme of section: 1—hopper; 2—adjustable bulkhead; 3—InGaS NIR cameras; 4—Multispectral VIS cameras; 5—pneumatic ejector; 6—product vessel; 7—contaminant vessel. (**b**) image VIS camera, (**c**) image NIR camera.

**Table 1 foods-13-00404-t001:** List of the correct classification rates for the datasets considered using nine classifiers.

Classifiers/Dataset	Broad Bean–Wheat	Broad Bean–Oat	Lentil–Wheat	Lentil–Oat	Chickpea–Wheat	Chickpea–Oat
	Correct classification rate (%)					
Decision trees	96.4	96.3	96.4	96.7	96.5	96.4
Discriminant analysis	94.3	94.3	94.6	94.5	94.8	94.5
Logistic regression	95.2	95.3	95.7	95.2	95.2	95.1
Naïve Baies	93.6	93.5	93.2	93.4	93.8	93.2
Support vector machine	100	100	100	100	100	100
Nearest neighbour	97.3	97.0	97.7	97.3	97.6	97.4
Kernel approximation	93.1	92.9	92.7	93.9	91.7	93.4
Ensemble	89.4	88.7	89.3	88.3	88.1	88.2
Neural network	98.2	99.1	99.2	98.1	98.2	98.0

**Table 2 foods-13-00404-t002:** Classification results based on the validation dataset.

Dataset	Wavelengths (nm)	% CCR	TPR	FPR	FNR
Broad bean–wheat	810; 1455	100.00	100.00	0.00	0.00
Broad bean–oat	655; 810; 1190; 530	100.00	100.00	0.00	0.00
Lentil–wheat	615; 1015	100.00	100.00	0.00	0.00
Lentil–oat	575; 1265	100.00	100.00	0.00	0.00
Chickpea–wheat	1595; 670; 1100	100.00	100.00	0.00	0.00
Chickpea–oat	1510; 845; 1075; 1660; 1310	99.20	99.20	0.00	0.80

CCR = correct classification rate; TPR = true positive rate; FPR = false positive rate; FNR = false negative rate.

**Table 3 foods-13-00404-t003:** Classification results based on the test dataset.

Dataset	Wavelengths (nm)	CCR (%)	TPR (%)	FPR (%)	FNR (%)
Broad bean–wheat	810; 1455	99.50	99.50	0.00	0.50
Broad bean–oat	655; 810; 1190; 530	99.11	99.11	0.00	0.89
Lentil–wheat	615; 1015	99.13	99.13	0.00	0.87
Lentil–oat	575; 1265	99.25	99.25	0.00	0.75
Chickpea–wheat	1595; 670; 1100	99.47	99.47	0.00	0.53
Chickpea–oat	1510; 845; 1075; 1660; 1310	99.14	99.14	0.00	0.86

CCR = correct classification rate; TPR = true positive rate; FPR = false positive rate; FNR = false negative rate.

**Table 4 foods-13-00404-t004:** Classification results using an industrial optical sorter.

Dataset	% CCR	% TPR	% FPR	% FNR
Broad bean—contaminant—run 1	99.44	99.44	0.00	0.56
Broad bean—contaminant—run 2	99.46	99.46	0.00	0.54
Broad bean—contaminant—run 3	99.46	99.46	0.00	0.54
Broad bean—contaminant—run 4	99.41	99.41	0.00	0.59
Broad bean—contaminant—run 5	99.42	99.42	0.00	0.58
Broad bean—contaminant—average	99.44	99.44	0.00	0.56
Lentil—contaminant—run 1	99.70	99.70	0.00	0.30
Lentil—contaminant—run 2	99.66	99.66	0.00	0.34
Lentil—contaminant—run 3	99.67	99.67	0.00	0.33
Lentil—contaminant—run 4	99.67	99.67	0.00	0.33
Lentil—contaminant—run 5	99.67	99.67	0.00	0.33
Lentil—contaminant—average	99.67	99.67	0.00	0.33
Chickpea—contaminant—run 1	99.34	99.34	0.00	0.66
Chickpea—contaminant—run 2	99.38	99.38	0.00	0.62
Chickpea—contaminant—run 3	99.35	99.35	0.00	0.65
Chickpea—contaminant—run 4	99.37	99.37	0.00	0.63
Chickpea—contaminant—run 5	99.37	99.37	0.00	0.63
Chickpea—contaminant—average	99.36	99.36	0.00	0.64

CCR = correct classification rate; TPR = true positive rate; FPR = false positive rate; FNR = false negative rate.

## Data Availability

Data is contained within the article.

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
