# Peer review of "Optimisation of an Industrial Optical Sorter of Legumes for Gluten-Free Production Using Hyperspectral Imaging Techniques"

_foods, 2024, doi:10.3390/foods13030404_

Round 1
Reviewer 1 Report
Comments and Suggestions for Authors
The researchers developed a sorting system for the grains, in terms of their gluten presence. It is a very important subject, but the manuscript is missing critical points that I have mentioned below.
· An abstract of a research paper should be concise and brief. But in this manuscript the authors included too much information, especially as a background. Also, the word limit of an abstract in journal of foods is 200. I recommend focusing on providing more detailed information about the methods employed, key results obtained, and major conclusions drawn, ensuring a more brief and informative representation within the constrained word limit.
· Line 50: the phrase “Due the growing” should be revised to “Due to the growing”. This example highlights one instance of a typo; however, many other typos and grammatical errors exist in the manuscript that need to through review and meticulous correction.
· The coherence between paragraphs needs improvement; better connections are required for a smoother flow.
· Have you employed any reference analysis to quantify the actual gluten content in the samples? What is the gluten content in the tested samples? Even if you are only predicting presence or absence of gluten, how you are testing the limits of your model? Is your model capable of detecting trace amounts of gluten, or does it become overwhelmed when confronted with samples containing too much gluten?
· Figure 2. It is better to show how you are actually testing the samples, instead of just showing the equipment.
· The results and discussion section requires improvement, specifically there is insufficient elaboration in the current discussion.
· Also, there is not literature comparison in the results section. How do your findings align with the existing literature?
Comments on the Quality of English LanguageModerate English editing is required.
Author Response
Thank you very much for taking the time to review this manuscript. Please find the detailed responses below and the corresponding revisions/corrections highlighted/in track changes in the re-submitted files.

Reviewer 2 Report
Comments and Suggestions for Authors
foods-2828905-peer-review-v1
Abstract:
The abstract delves extensively into gluten sensitivity, celiac disease, and the optical sorting system for contaminant-free legume production. However, it requires refinement to fulfill its purpose more effectively. There's a need for a clearer distinction between the severity of gluten sensitivity and celiac disease to avoid potential confusion among readers. While the research objective is well-stated, earlier emphasis on the optical sorting system's significance in identifying spectral characteristics for contaminant-free legume production would amplify its impact. Although informative, the methodology description needs better emphasis on the role of data complexity reduction in handling large datasets. Providing specific details about the system's efficiency in removing contaminants will notably enhance comprehension. Moreover, expanding its focus beyond production chains to showcase broader practical applications will heighten its relevance. On the other hand, the abstract contains excessive introductory content better suited for the manuscript's introduction. It lacks specific details regarding the applied methodology, results, and key conclusions, thereby overlooking crucial aspects. A comprehensive overhaul is necessary to ensure conciseness, effectively encapsulating the study's purpose, introducing the topic, detailing the methodology, presenting the findings, and concluding succinctly.
Introduction:
The introduction is quite comprehensive, covering various facets related to gluten sensitivity, celiac disease, and the challenges of identifying gluten contaminants in gluten-free products. However, it might benefit from a more streamlined approach to improve its effectiveness.
· Structure and Flow: The paragraph transitions between different aspects—ranging from disease prevalence to detection methods to innovative solutions—without clear delineation. Segmenting this information into distinct sections would enhance readability and coherence.
· Objective Clarity: While the paragraph delineates various issues related to gluten contamination and the limitations of current detection methods, it lacks a clear, upfront statement about the primary objective or aim of the study. Clearly articulating the study's focal point in detecting gluten contaminants using hyperspectral imaging would help orient the reader.
· Specificity and Relevance: While discussing the shortcomings of existing methodologies, linking these limitations to the relevance and importance of hyperspectral imaging within this study's context could provide more context and focus.
· Conciseness: The paragraph tends to delve deeply into methodological aspects and past studies, potentially overwhelming the reader with details. Condensing these details without sacrificing essential information would enhance readability.
· Integration of Citations: The citations are scattered across the text, sometimes disrupting the flow. Integrating them more seamlessly or consolidating related citations into distinct sections could enhance the paragraph's coherence.
Material and Methods:
The "Materials and Methods" section provides a comprehensive overview of the experimental processes, tools, and analytical methods used in the research. It delivers detailed information about the samples, instruments, and analytical techniques utilized, although there are some ambiguities and shortcomings that need addressing for better reader comprehension. Areas for Improvement:
· Symbol Notations: Address inconsistencies in symbol notations (lines 163-167) to ensure clarity and consistency throughout the text.
· Equation Descriptions: Below each equation, provide comprehensive explanations of all variables and constants involved, along with their respective units of measurement, for better reader comprehension.
· Preliminary Results: Including preliminary results (lines 172-173) as an appendix would enhance the manuscript, offering readers deeper insights into the research.
· Graph Clarity: Improve graph titles to ensure standalone interpretability by specifying and describing all abbreviations, symbols, and physical quantities presented on them. This will aid readers in comprehending the graphs without relying on the accompanying text.
· Camera Details: Detailed information about the cameras used (line 220) is necessary to enhance the understanding of the equipment utilized for the research.
Results:
· Inadequate Interpretation: The section lacks detailed interpretation of the significance of obtained results, especially regarding the practical implications of classification rates and error percentages.
· Inconsistencies in Terminology: The use of inconsistent terminologies and descriptions across tables creates confusion and difficulty in interpreting and comparing data. For instance:
o Varying Labels: Labels and notations in the tables are inconsistent across different datasets, making it challenging to correlate the data. This lack of uniformity in labeling hinders the reader's ability to understand the relationships between variables and datasets.
o Measurements Discrepancies: There are discrepancies in measurements provided across different tables. For instance, wavelengths are represented differently in various datasets, impeding straightforward comparisons and data synthesis.
Recommendation: Ensure uniformity in the presentation of data labels and measurements across all tables. Adopt standardized notation and units for variables and measurements to facilitate easy cross-referencing and comprehension.
· Absence of Contextual Analysis: The discussion fails to contextualize the observed differences in error rates among various datasets or runs. For example:
o Lack of Explanations: While the results exhibit varying error rates among datasets or runs, the discussion does not offer explanations for these differences. It's crucial to provide contextual analysis, elucidating why certain datasets or runs displayed distinct error rates.
o Factors Influencing Variability: Factors influencing the observed variations in error rates (such as environmental conditions, sample characteristics, or measurement variations) need to be elucidated. This contextual information aids in understanding the reliability and consistency of the results.
Recommendation: Provide a detailed contextual analysis discussing potential factors contributing to the variability in error rates across different datasets or runs. Clarify the reasons behind the observed differences to offer a comprehensive understanding of the results.
· Missing Data Explanation: The manuscript lacks explicit details regarding the reasons behind the error percentages and classification rates, specifically concerning false negatives in the industrial tests:
o Lack of Clarity: The absence of a clear explanation or elucidation regarding the reasons behind certain error rates, particularly false negatives in the industrial tests, creates ambiguity. This omission limits the readers' understanding of why specific errors occurred.
o Insufficient Insight: Without a detailed explanation, readers are left without insights into the circumstances or conditions leading to these errors. Understanding these factors is crucial for the reliability and generalizability of the results.
Recommendation: Include a thorough explanation or discussion section elucidating the reasons and circumstances contributing to observed error rates, especially false negatives in the industrial tests. This will provide clarity and enhance the credibility of the results.
· Limited Discussion of Industrial Constraints: Insufficient discussion regarding the constraints of the industrial optical sorter's real-time application, especially regarding product flow speed and orientation issues impacting the false negative rates.
Conclusions:
· Lack of Recommendations: The section does not offer clear recommendations for resolving identified issues or improving the classification system's efficiency.
· Incomplete Solution Proposals: Although the text mentions potential areas for improvement (e.g., product conveying system enhancements), there is a lack of comprehensive suggestions or strategies to address these shortcomings.
· Missing Future Research Directions: It lacks a clear outline of future research directions or areas for further investigation to overcome the identified limitations.
Comments on the Quality of English LanguageThe language in the manuscript is generally clear and comprehensible. However, there are instances where the structure of sentences could be improved for better readability.
Author Response

(The authors gave the same response as above.)
